# The Effects of Temperature on Accident and Emergency Department Attendances in London: A Time-Series Regression Analysis

**DOI:** 10.3390/ijerph17061957

**Published:** 2020-03-17

**Authors:** Ines Corcuera Hotz, Shakoor Hajat

**Affiliations:** 1Faculty of Public Health and Policy, London School of Hygiene and Tropical Medicine, London WC1H 9SH, UK; 2Department of Public Health, Environments and Society, Centre on Climate Change and Planetary Health, London School of Hygiene & Tropical Medicine, 15-17 Tavistock Place, London WC1H 9SH, UK; shakoor.hajat@lshtm.ac.uk

**Keywords:** temperature, weather, climate, emergency department, attendances, time-series

## Abstract

The epidemiological research relating mortality and hospital admissions to ambient temperature is well established. However, less is known about the effect temperature has on Accident and Emergency (A&E) department attendances. Time-series regression analyses were conducted to investigate the effect of temperature for a range of cause- and age-specific attendances in Greater London (LD) between 2007 to 2012. A seasonally adjusted Poisson regression model was used to estimate the percent change in daily attendances per 1 °C increase in temperature. The risk of overall attendance increased by 1.0% (95% CI 0.8, 1.4) for all ages and 1.4% (1.2, 1.5) among 0- to 15-year-olds. A smaller but significant increase in risk was found for cardiac, respiratory, cerebrovascular and psychiatric presentations. Importantly, for fracture-related attendances, the risk rose by 1.1% (0.7, 1.5) per 1 °C increase in temperature above the identified temperature threshold of 16 °C, with the highest increase of 2.1% (1.5, 3.0) seen among 0- to 15-year-olds. There is a positive association between increasing temperatures and A&E department attendance, with the risk appearing highest in children and the most deprived areas. A&E departments are vulnerable to increased demand during hot weather and therefore need to be adequately prepared to address associated health risks posed by climate change.

## 1. Introduction

It is well established that heatwaves can affect health [1,2]. Since Europe’s severe heatwave in 2003 [3], the health impact of weather has received increasingly more attention. The Intergovernmental Panel on Climate Change reported in 2018 that human activities have caused approximately 1.0 °C of global warming above pre-industrial levels, and this is likely to reach 1.5 °C above pre-industrial levels between 2030 and 2052 if it continues to increase at the current rate [4]. More recent evaluations of the progress towards the Paris Agreement have shown that the current policies presently in place around the world are projected to reduce baseline emissions and result in 3.0 °C of warming above pre-industrial levels [5]. In all realistic scenarios of climate change, the frequency and intensity of extreme weather events is likely to increase [6], raising the question of how best to meet the associated rise in demand to front-door health services.

Numerous studies have described the non-linear and delayed relationship between mortality and ambient temperature worldwide [6,7,8,9]. Typically, the relation is U-,V- or J- shaped, where the risk of death may increase at both very low and very high temperatures [10,11,12]. While the effects on mortality and hospital admissions have been well researched, few have investigated the impact of changes in temperature on Accident and Emergency (A&E) department attendances. In Taiwan, a time-series study was used to investigate the effects of meteorological factors on A&E department revenues [13]. The authors found a positive correlation between maximum daily temperature and non-trauma attendance. Studies from Australia, China, Singapore and Brazil have associated daily patient attendances with factors such as public holidays, precipitation, temperature, air pollution and relative humidity [8,14,15,16,17]. In the United Kingdom (UK), only a few studies have examined the effect of temperature on overall and multi-cause attendance rates. Most studies have focused on a single cause of attendance such as trauma, fractures or assault and found that high temperatures are significantly and positively correlated with the number of presentations [18,19].

This study aims to investigate the association between ambient temperature and the occurrence of daily A&E department attendances in different age groups living in Greater London (GL). Due to the higher density of buildings and dark surfaces and less vegetation, GL has a microclimate known as “urban heat island” [20]. With greenhouse gas emissions rising at unprecedented levels and leading to an increase in mean temperatures, the urban heat island effect is likely to become even more prominent [21,22]. With additional challenges such as a growing urban population, patients attending with complex multimorbidity and ongoing healthcare staff shortages, the need to quantify the impact ambient temperature has on emergency attendances is important. In this study, we report the effects of temperature on overall and cause-specific attendances at different lag periods. Additionally, age-groups and deprivation quintiles are investigated as potential effect-modifiers of the temperature effect.

## 2. Materials and Methods

### 2.1. Data

Data on all A&E department attendances among GL residents for the period from 1 April 2007 to 31 March 2012 were obtained from a national database (NHS Digital). Primary outcomes were defined as daily A&E department attendance counts that were age-specific to 0–15, 16–64, 65–74, 75–84 and 85+ years, and cause-specific for the following diagnostic categories: “cardiac conditions”, “cerebrovascular conditions”, “respiratory conditions”, “psychiatric conditions”, “fractures” and “social conditions” (the latter including alcoholism and homelessness). These categories were predefined by the A&E NHS Digital data dictionary, a rich source of detailed records made up of individual records for all A&E department attendances (including deaths) occurring in England [23]. These items form part of the national Commissioning Data Set, and are generated by the patient administration systems within each hospital covering each financial year. The data assist in the production of aggregate summaries and thereby ensures patient confidentiality.

Weekly influenza counts were obtained from laboratory-confirmed influenza notifications made to Public Health England’s Communicable Disease Surveillance Centre.

Deprivation levels were considered as potential effect-modifiers of the temperature effect, and were measured using the English Index of Multiple Deprivation (IMD). The IMD, aggregated at the Super Output Area level, combines information from several domains such as income, employment, education, housing, and health, to produce an overall measure. IMD values were collapsed into quintiles, with “Quintile 1” being the most deprived and “Quintile 5” the least deprived, and subsequently linked to the A&E department data.

Daily meteorological data were measured by 31 weather stations within the GL region and obtained with permission from the Medical and Environmental Data Mash-up Infrastructure database (MEDMI) [22]. Hourly observed air temperature and relative humidity data were used to create daily series for the whole region. Daily mean temperatures were estimated by averaging daily maximum and daily minimum values.

Daily air pollution levels, particulate matter of less than 10 micrometers in diameter (PM_10_) and ozone (both measured in μg/m^3^) were derived from one Automatic Urban and Rural Network (AURN) monitoring site in central London (Bloomsbury station).

### 2.2. Statistical Methods

The relation between daily A&E department attendances and daily mean temperature was investigated using Poisson generalized linear models adjusted for autocorrelation and overdispersion. Natural cubic splines of date were used to control for secular trends and any additional confounding by seasonally varying factors other than temperature. Sensitivity analyses using varying seasonal control, ranging from 7 to 12 degrees of freedom (df) per year, confirmed that the model with 12 df per year showed random dispersion of points in the residual plot as well as the lowest Akaike Information Criterion (AIC), and was therefore chosen for subsequent analyses. This allowed adequate control of unmeasured confounders while leaving sufficient information from which to estimate the temperature effect. All models included indicator terms to adjust for the effects of day of the week and UK public holidays. In addition, daily measures of same-day humidity were incorporated in the core regression model using a natural cubic spline with 5 df, as well as weekly counts of influenza. As the effect of air pollution on heat is routinely considered in heat–mortality studies [7,12], we controlled for ozone and PM_10_ during the sensitivity analysis stage to check the robustness of the results. To detect autocorrelation, model diagnostic tests were run such as the minimization of the partial autocorrelation function. If autocorrelation was detected it was controlled for by lagging the residuals and adding it into the core regression model as an explanatory term.

The functional form of the temperature–attendance relationship was visually assessed by constructing graphs of attendances as smoothed functions of temperature using natural cubic splines with 4 df. To explore whether there was any delayed or “lagged” association between outcome today and exposure on previous days, the exposure series was shifted forward in time using two different lag structures. First, the relationship was modelled using temperature averaged across the same day and two days before the day of the A&E department visit (lags 0–2). Then, the effect was modelled using temperature averaged across values lagged over 21 days before the A&E department attendance (lags 0–21). These choices were based on previous research, showing that excess risk caused by heat is typically immediate [6,24] and occurs within a few days (therefore we used a time lag of 0–2 days), while the effects of cold have been reported to last up to a few weeks (time lag of 0–21 days used) [7,9,25].

To quantify the relationship for the final results, an unconstrained distributed lag model was used to assess the effect. Each lag was first modelled separately then summed for the total temperature effect on A&E department attendances. Threshold temperatures were identified by looking at each cause-specific attendance graph in turn. If visualized, they were incorporated into the model as a linear term.

The results of this analysis provide an estimate of A&E department attendances attributable to the weather over and above the seasonal and daily “norms” predicted from the baseline, and therefore the effect estimates are expressed as percentage changes in daily patient attendances.

All statistical analyses and figures were carried out in STATA (v15, StataCorp LLC, College Station, TX, USA) except Figures 3 and 4, which are from Excel.

## 3. Results

### 3.1. Descriptive Results

Emergency Department data, meteorological factors and air pollutants are summarized in Table 1. There were 13,400,000 attendances to GL A&E departments, with an average of 7349 per day, from 1 April 2007 to 31 March 2012. More than half of all patients attending A&E departments (53%) showed relatively high levels of social and economic deprivation as measured by the IMD (deprivation Quintiles 1 and 2). The daily mean temperature from June to August was 19.1 °C, and from December to February it was 5.8 °C. Relative humidity, ozone and PM_10_ had mean values of 71.40%, 25 µm/m^3^ and 22 µm/m^3^, respectively. PM_10_ and ozone were found to have missing values of 7.1% and 4.5%, respectively, of the total data.

Figure 1 shows raw plots of daily total A&E department attendances and daily mean temperature over time, indicating that both series are dominated by annual seasonal patterns, with peaks in the summer and troughs in the winter. A&E department attendances also showed an increasing trend over the long term, possibly due to an increasing number of hospitals providing their data, among other reasons.

### 3.2. Main Analysis

Figure 2a,b describes the fitted relation between daily mean temperature and daily A&E department attendances by cause. The curves show the relative risk for the number of attendances for values across the temperature range. The center line is the estimated spline curve, and the upper and lower lines represent the 95% upper and lower confidence limits, respectively. All six cause-specific outcomes were graphically inspected using both short (0–2 day) and long (0–21 day) lag structures to look for an effect of temperature. For overall attendances (Appendix A), the relation appears approximately flat across the entire temperature range, thus daily mean ambient temperature was added as a linear term to the final regression model for all outcomes except the fracture-related attendances (lagged at 0–2 days). The temperature threshold for fractures was visually determined at 16 °C and incorporated as a linear threshold term into the regression model. To formally test for evidence of non-linearity, the AIC was compared between the model with the temperature effect specified as a linear term and the model with the temperature threshold term. The model with the lowest AIC was subsequently chosen for the regression analysis. There was no evidence of a heat temperature threshold for other outcome-specific series.

No apparent effect of low temperature on any of the cause-specific A&E department attendances was seen; therefore, the cold effect was not modelled further.

Table 2 is a summary of the main findings from the regression model investigating the total (or summed) temperature effect by cause and by age. For ease of interpretation, the total effect of each lag estimate is displayed and expressed as percentage change in attendance for each 1 °C degree increase in daily mean temperature. The effects of individual lags are shown in more detail in Appendix A. The results for fracture-related attendances are displayed as the percent change in risk for A&E department attendance for each 1 °C above the identified threshold of 16 °C.

All causes, except attendances for social conditions, displayed statistically significant effects of rising temperature (at the 5% level). For all-cause attendances, the strongest effect was seen in the youngest age group (0–15 years) displayed in Figure 3, with an increase of 1.38% (95% CI 1.21–1.54). A pronounced effect of temperature was seen in those aged 65–74, where attendances related to respiratory conditions increased by an average of 1.54% (95% CI 0.91–2.19) for each degree increase in temperature. Cardiac and cerebrovascular conditions also showed a small but significant increase in response to increases in temperatures. The relation was mostly flat for social conditions. Fracture-related attendance rates increased by 2.1% (95% CI 1.5–3.0) among children aged 0- to 15-years for each 1 °C increase above the threshold of 16 °C.

### 3.3. Effect Modification by Deprivation

To identify vulnerable subpopulations, models were run separately for each outcome and deprivation quintile, and results are visually displayed for all-cause attendances in Figure 4 (numerical results are listed in Appendix A and all cause-specific outcomes are illustrated in Appendix A). Overall, these suggest a slight decrease in risk of A&E department attendance across the deprivation quintiles. For all-cause attendances, the most deprived quintile (Quintile 1) showed the highest risk of A&E department attendance (1.02%, 95% CI 0.93, 1.10) per 1 °C increase in temperature (Figure 4). Among the cause-specific attendances, the highest risk (1.82%, 95% CI 0.97, 2.66) was seen in the fracture-related attendances among the most deprived quintile (Appendix A).

## 4. Discussion

This ecological time-series regression study provides evidence of an effect of increasing temperature on daily A&E department attendances in GL, which persisted even after controlling for seasonality, long-term trend and day of the week. The percentage change of overall A&E department attendance increased for almost all cause-specific attendances, except for social conditions. Although the effect estimates may appear small, they are of public health concern since the entire population is exposed [26].

The association between high temperatures and an increase in hospital admissions for respiratory problems, renal disease [6] and injuries [27] have previously been described in high-income country settings. Atherton et al. found that pediatric but not adult trauma cases rose over the summer months in the UK, and the authors further found this to be correlated with a rise in daily temperatures, more hours of sunshine and fewer millimeters of rainfall [28]. Our study found the average temperature above which heat-related A&E department attendances increased due to fractures to be at 16 °C. The greatest effect was seen among the 0- to 15-year-olds (2.1% 95% CI 1.5–3.0). Previous studies have found similar effects in UK pediatric populations. Several authors suggest that longer hours of sunshine lead to more outdoor activity and an increase in injuries and falls, increasing the overall risk of fractures [18,28]. This effect appears to be more indirect rather than a direct physiological response to heat. However, a recent systematic review analyzing the impact of high ambient temperature on unintentional injuries in all ages found an increase in injuries in relation to high temperatures in 11 out of 13 studies [27]. When analyzing the effects by age group, the authors concluded that the evidence is still limited and at times contradictory. The most common cause cited for the observed relationship were changes in behavior, meaning more outdoor activity in warmer months, in keeping with the above assumption [27]. The effect of temperature on adults attending with fractures was much smaller, and for the oldest age group (75- to 84-year-olds) the attendance rate even fell above the chosen temperature threshold. Previous studies suggest that fracture rates in this age group increase in the winter due to cold and falls on snow and ice, although findings are highly variable between studies [19,28,29].

Xu et al. [14] analyzed the effects of extreme temperatures on pediatric A&E department admissions in Australia and found that children were particularly vulnerable to the effects of high but also low temperatures, effects that were seen across a range of pediatric diseases. A&E department attendances are distinct from emergency admissions, however. According to a report of the House of Commons Health committee “attendances peak during the summer months but hospitals experience most pressure and struggle hardest to achieve the four-hour waiting time standard during the winter” [30]. This is thought to be due to a greater proportion of people attending A&E department s in the winter who require emergency admission. Overall in the UK, A&E department attendances show an average admission rate that varies from 27.9% in the winter compared to 25.8% in the summer [30].

None of the six outcomes in this study displayed any significant cold effect. This was investigated visually through various models using natural cubic splines, but the results suggested a fairly linear relationship throughout. Several combinations of lag days were assessed in the sensitivity analysis (14-day lag and 21-day lag), but the relationship remained mostly flat. Other international studies found a U-shaped relationship between temperature and A&E department attendances similar to that seen with mortality data [8,15]. A study conducted in Southern New England demonstrated that both high and low temperatures were associated with higher rates of all-cause A&E department attendances and that the slope of the exposure-response function was steeper for high temperatures than for low temperatures [8]. So far, no study has been conducted in London on the effects of weather on A&E department attendances, but a detailed London time-series study on ambulance call-out incidence and response time showed that as temperature rises above 20 °C and falls below 2 °C, the total ambulance call-out volume increases [31]. Furthermore, ambulance response times were proportionally more affected by cold weather due to slippery roads from ice and snow, whereas, in warm temperatures, the roads and traffic were not normally affected. This shows that changes in temperature do not have to be severe for an impact on ambulance services to occur in the GL area [31].

Temperature was associated positively with all cause-specific A&E department attendances, except for social conditions. Given the complex and multifactorial nature of these presentations, temperature alone may not play a significant role in A&E department attendance. Another reason for not finding an association could be that alcohol may not always be the primary problem that leads to an A&E department attendance, but rather the contributing element in, for example, a fracture or cardiovascular presentation. Data coding may have been affected by information bias, making the outcome series “noisier” and thereby biasing the effect estimates towards the null. A recent survey of emergency department consultants found that alcohol-related incidents accounted for 25% of the caseload [30]. Only a few studies looked at utilization rates by homeless patients and the association to daily temperature. A retrospective study conducted in Sheffield did not find evidence to suggest that homeless people are more likely to attend the emergency department in cold weather [32].

Finally, when investigating for possible effect modification by deprivation quintile, graphical inspection appeared to suggest that those in the most deprived quintiles are at higher risk of attending an A&E department compared to the least deprived quintiles. Deprived populations are likely to be more susceptible or less adapted to the heat (due to less thermally efficient housing or no availability of air conditioning), and therefore more exposed to higher temperatures and less likely to recover from illnesses triggered by heat. Population health and social as well as environmental inequalities vary greatly across GL, and consequently may affect attendance rates differently across different A&E departments.

This study had several limitations. Firstly, ozone and PM_10_ were derived from one AURN monitoring site in central London, which may not be representative of all GL localities. Secondly, the data sets for air pollution had 4.5% days of missing observations for ozone and 7.1% for PM_10_. To address this issue, the variables for air pollution were added at the sensitivity analysis stage, but no considerable difference was noted in the final effect estimates. While air pollution effects were not the main focus of this study, we believe that it is important to consider how heat effects are potentially impacted by air pollution control as they are routinely considered in heat-mortality data [12,33]. Thirdly, risk assessment in environmental epidemiology was challenged by the complexity of accurately measuring the exposure to temperature [34]. Environmental variables such as temperature were often available at population level and assumed to be equal across GL. However, the risk to health from extreme temperatures is dependent on many individual factors such as personal behavior and indoor vs outdoor exposure, as well as housing design and the general built environment [22]. This study was unable to take account of adaptive measures (such as air conditioning) or population acclimatization, which could have introduced bias through the misclassification of the exposure.

## 5. Conclusions

This study aimed to establish and quantify the short-term association between temperature and risk of attending A&E departments in Greater London using time-series regression analyses adjusted for multiple confounders. There was a positive association between temperature and A&E department attendance, with the most pronounced effect seen in the youngest age group and the most deprived quintiles. Although understanding the impact weather has on attendance rates is an important first step for devising adaptation and mitigation strategies, this information alone may not be enough for the strategic planning of an A&E department. Further research in this field is needed to increase the evidence, feasibility and cost-effectiveness of employing staff at short notice following forecasts of hot weather. However, A&E departments are key locations for surges in demand following heat exposures, and require special attention in national heat warning systems. More awareness of summertime surges in demand is also needed among hospital staff, a time when clinicians and hospital managers often expect a period of respite from the ever-growing pressures facing emergency departments.

## Figures and Tables

**Figure 1 ijerph-17-01957-f001:**
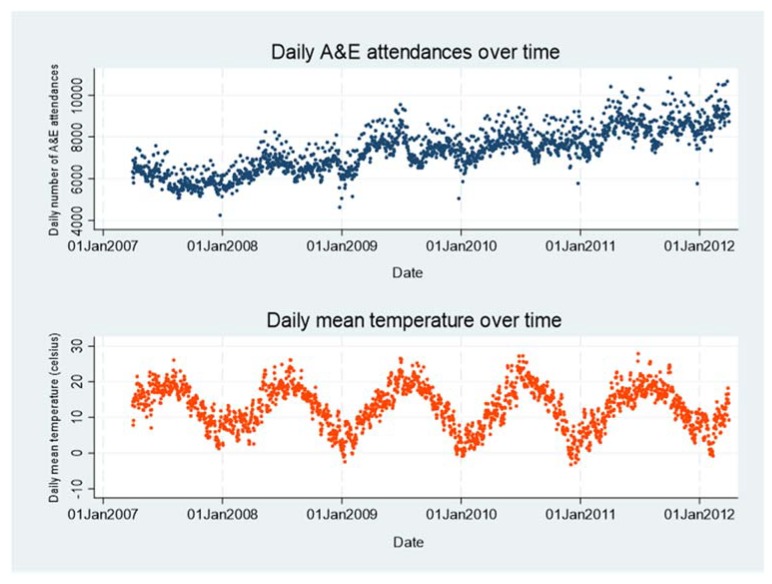
Raw plots showing Accident & Emergency (A&E) department attendances and mean temperature data over the five-year study period in Greater London.

**Figure 2 ijerph-17-01957-f002:**
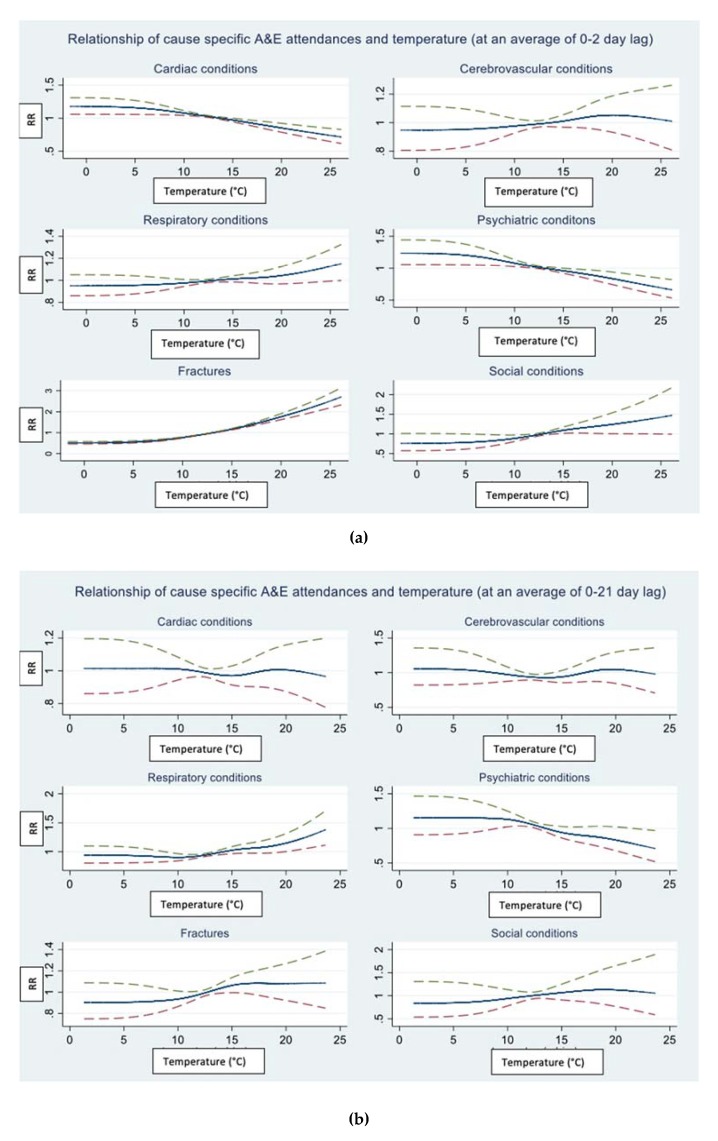
Relations between daily mean temperature and A&E department attendance by cause. The model is adjusted for seasonality, trend and day of the week. Temperature (˚C) on the x-axis modelled with (**a**) 0- to 2-day lag and (**b**) 0- to 21-day lag and relative risk (RR) on the y-axis.

**Figure 3 ijerph-17-01957-f003:**
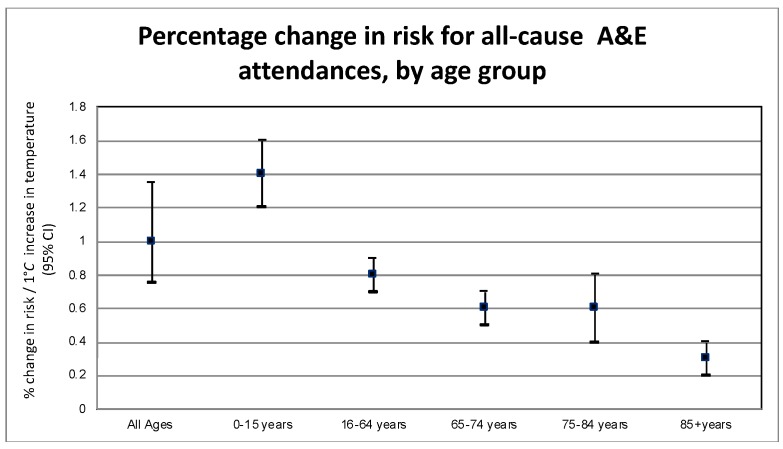
Risk of attending an A&E department per 1 °C increase in mean temperature for all causes, by age group.

**Figure 4 ijerph-17-01957-f004:**
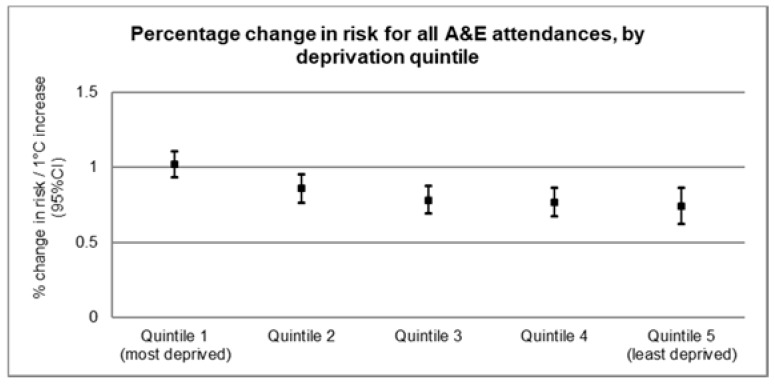
Effect modification by deprivation quintile for all-cause A&E department attendances in Greater London between 2007 and 2012.

**Table 1 ijerph-17-01957-t001:** Descriptive statistics of the Emergency Department data, meteorological factors and air pollutants in Greater London (GL) between 2007 and 2012.

Variables		Mean	SD	Min	25th–75th (Percentiles)		Max
*Age Group*							
	<15 years	1405.5	280.2	733	1208	1601	2272
	16–64 years	4522.4	695.6	2437	3988	5017	6932
	65–74 years	445.7	75.6	224	349	499	671
	75–84 years	430.5	71.9	228	377	484	621
	85+ years	262.7	47.3	135	228	297	437
	All ages	7349.1	1077.4	4248	6505	8124	10,849
*Deprivation*							
	Quintile 1	1524	208.9	949	1367	1669	2187
	Quintile 2	2289	397.2	1278	1974	2582	3556
	Quintile 3	1631	239.8	915	1456	1789	2380
	Quintile 4	1144	165.0	713	1027	1251	1732
	Quintile 5	687	99.4	360	610	759	1018
*Cause-specific attendances*							
	Cardiac	95.5	18.3	43	82	108	162
	Respiratory	175.4	52.9	79	135	202	478
	Fractures	172.3	35.1	63	148	194	353
	Cerebrovascular	37.6	13.2	11	27	79	88
	Psychiatric	35.9	8.3	11	30	42	62
*Environmental variables*							
Mean Temperature (°C)		12.80	5.99	−3.19	8.45	17.42	27.90
	PM_10_ * (µm/m^3^)	22.05	11.51	6	14	27	89
	Ozone (µm/m^3^)	25.47	15.42	0	13	36	81
Relative Humidity (%)		71.38	13.68	35.86	60.38	82.07	98.90

* particulate matter of less than 10 micrometers in diameter.

**Table 2 ijerph-17-01957-t002:** Summary of main findings from the regression model investigating the heat effect by cause and by age. Results shown are for the total effect lagged over 0–2 days.

Heat Related % Change in Risk of A&E Department Attendance Per 1 °C Increase in Mean Temperature (95% CI^1^)		
*1.* *All Attendances*		
0–15 years	1.4	(1.2, 1.6)
16–64 years	0.8	(0.7, 0.9)
65–74 years	0.6	(0.5, 0.8)
75–84 years	0.6	(0.4, 0.7)
85+ years	0.3	(0.2, 0.5)
all ages	1.0	(0.8, 1.4)
*2.* *Cardiac*		
16–64 years	0.7	(0.4, 1.1)
65–74 years	0.6	(0.1, 1.3)
75–84 years	0.7	(0.1, 1.3)
all ages	0.7	(0.4, 1.0)
*3.* *Respiratory*		
0–15 years	0.6	(0.1, 1,0)
16–64 years	0.4	(−0.1, 0.7)
65–74 years	1.5	(0.9, 2.2)
75–84 years	0.7	(0.1, 1.3)
all ages	0.7	(0.4, 0.9)
*4.* *Cerebrovascular*		
16–64 years	1.0	(0.5, 1.6)
all ages	0.6	(0.2, 1.0)
*5.* *Psychiatric*		
16–64 years	0.7	(0.3, 1.1)
all ages	0.7	(0.3, 1.0)
*6.* *Social*		
all ages	0.4	(−0.3, 1.1)
*7.* *Fracture attendances for each 1 ˚C increase above the threshold of 16 ˚C*		
0–15 years	2.1	(1.5, 3.0)
16–64 years	0.9	(0.3, 1.4)
65–74 years	0.9	(−0.4, 2.2)
75–84 years	−0.4	(−1.7, 0.9)
all ages	1.1	(0.7, 1.5)

(1) Abbreviations: CI: confidence interval. (2) The unconstrained distributed lag model controls for seasonality and trend as well as day of week, public holidays, relative humidity and influenza. Results show the combined effect of 0- to 2-day lag. (3) Cause-specific age subgroups that had less than 10 daily counts were not included in the analysis due to lack of power.

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
