# Peer review of "The Effects of Temperature on Accident and Emergency Department Attendances in London: A Time-Series Regression Analysis"

_ijerph, 2020, doi:10.3390/ijerph17061957_

Round 1

Reviewer 1 Report

Dear Editor of the International Journal of Environmental Research and Public Health

The article "The effects of temperature on the London Accident and Emergency Department's undertakings: an analysis of time series" has been revised.

Some important considerations and notes are necessary in the text, which are indicated below:

In the introduction part of the text, the authors make an interesting demonstration about the problem of climatic variability and the incidence of medical records in the city of London. It is recommended that the main objective of the study be defined more clearly at the end of the INTRODUCTION.

In line 25 of the ABSTRACT, it is suggested to change a statement "optimize the search, quality and safety of care".
The optimization of the health care service presupposes pressure and a greater demand for results from people assisted, or it can lead the medical team to errors, procedures and inadequate diagnoses of patients. Be careful what you write!

Reference number 19, despite being a review study on the heat island, is not the most adequate to faithfully represent the study on the phenomenon of the urban heat island. It is highly recommended that the work of Professor OKE be referenced.

It is highly recommended that the authors of the article rewrite part of the text and remove the data on air and ozone pollution from the study, as they have lost scientific rigor due to the deficiency and absence of primary data.

In addition, the study will lose its quality due to the two statements between lines 87-89
". Due to the lack of 88 data, air pollution was not included in the main model, but added during the sensitivity analysis stage 89 to verify the robustness of the results"
and lines 265-270
"This study has several limitations. First, ozone and pm10 were derived from just one AURN monitoring site in central London, which may not be representative of all GL locations. Second, the 266 data sets for air pollution had 83 missing data points (4.5%) for ozone and 129 missing data points (7.1%) for pm10 out of a total of 1,827 data points. It was not possible to determine whether air pollution data air were randomly absent or if there was a systematic reason for this, potentially creating bias in the results.To solve this problem, variables for air pollution were added at the sensitivity analysis stage, but no considerable difference was observed in the final estimates of the effect ".

This reviewer appreciates the opportunity to collaborate with the International Journal of Environmental Research and Public Health.

Author Response

Dear Reviewer of the International Journal of Environmental Research and Public Health,

Many thanks for your feedback and comments on the article "The effects of temperature on London Accident and Emergency Department attendances: a time-series regression analysis".

Please find below my point-by-point responses below:

POINT 1. It is recommended that the main objective of the study be defined more clearly at the end of the INTRODUCTION.

The aim of this study is now clearly defined in the last paragraph of the introduction (p.2 l.51-61).

POINT 2. In line 25 of the ABSTRACT, it is suggested to change a statement "optimize the search, quality and safety of care".

Line 24-25 amended to remove the idea of adding additional pressure on the health care service providers.

POINT 3. Reference number 19, despite being a review study on the heat island, is not the most adequate to faithfully represent the study on the phenomenon of the urban heat island. It is highly recommended that the work of Professor OKE be referenced.

Many thanks for pointing this out. OKE has now referenced in his works from 1982 where he describes the Urban Heat Island effect.

POINT 4. It is highly recommended that the authors of the article rewrite part of the text and remove the data on air and ozone pollution from the study, as they have lost scientific rigor due to the deficiency and absence of primary data. In addition, the study will lose its quality due to the two statements between lines 87-89 and lines 265-270.

Although air pollution effects were not the main focus of this study, we believe that it is important to consider how heat effects were potentially impacted by air pollution control, as is routinely done in heat-mortality papers. We are reassured that control for O3 and PM10 made little difference to our estimates. However, as suggested by the author, we have reworded some of the text (page10 line 260 -262) to better reflect the fact that the quality of the available air pollution data was not a major factor in the findings of this study.

Reviewer 2 Report

This study examined the associations between temperature and ED visits in London from 2007 to 2012. The meteorology data used in this study were measured by multiple weather stations, and the data of ED visits were obtained from all ED departments in London. The study has a large sample size and the study population is representative of a large population. Statistic methods are well designed in this study. Specific comments are here:

Page 62-63, the categories of age groups. The interval of different age groups is different. Especially, the 16-64 group is much wider than the 0-15 or 65-74. The author should provide a justification. Line 72. I’m not sure if the daily mean temperature should be estimated using the daytime temperature. It seems the night time temperature or daily variation is also important to people’s health. The author may need to provide a reference to taking the average from 9am to 9pm. Line 111-112. It is related to my first comment. The high number of ED visits in the 16-64 age group may be due to that this group has more people than other age groups. The author can change the percentage to rate. Line 115. I think the confidence interval should not be used in describing variations. Confidence interval means how much confidence that the true association is located in this range. But these sentences are not talking about associations. The author may change the 95%CI to percentile to describe the variation of those study variables. Table 1. Why gender and race are not adjusted in the model? Figure 2. The labels are missing for all y-axis. Results: The 0-15 age group shows a significant association between temperature and ED visits. However, it may be due to that this age group has increased activity and high risk of injury during summer. If that, the temperature should not be the cause of those ED visits. The author needs to make a discussion in the manuscript. Line 235. I think this result is reasonable because the effect of cold temperature is more common in low attitude areas.

Author Response

Dear Reviewer of the International Journal of Environmental Research and Public Health,

Many thanks for your feedback and comments on the article "The effects of temperature on London Accident and Emergency Department attendances: a time-series regression analysis".

Please find below my point-by-point responses to your comments:

POINT 1: line 66, the categories of age groups. The interval of different age groups is different. Especially, the 16-64 group is much wider than the 0-15 or 65-74. The author should provide a justification.

The age-groupings were based on bands commonly used in the literature and chosen to broadly reflect children, adults, elderly and the very elderly. Having equal numbers in each age-group was not a consideration.

POINT 2: Line 72. I’m not sure if the daily mean temperature should be estimated using the daytime temperature. It seems the night time temperature or daily variation is also important to people’s health. The author may need to provide a reference to taking the average from 9am to 9pm.

Apologies this was an error in wording. The daily mean temperature was derived by averaging daily maximum and daily minimum values. This better reflects both day- and night-time exposures, both of which may be important to health as the reviewer suggests. We have clarified this in the revised text (line 80, page 2).

POINT 3: Line 111-112. It is related to my first comment. The high number of ED visits in the 16-64 age group may be due to that this group has more people than other age groups. The author can change the percentage to rate.

We agree that this will reflect greater numbers in this particular age-group, however since we are interested in relative heat effects within each age band, we feel it is better to simply omit the superfluous sentence rather than introduce rates (line 120 , page 3).

POINT 4: Line 115. I think the confidence interval should not be used in describing variations. Confidence interval means how much confidence that the true association is located in this range. But these sentences are not talking about associations. The author may change the 95%CI to percentile to describe the variation of those study variables.

We agree with this suggestion, confidence intervals have been removed (line 118-124, page 3). Percentiles are displayed in Table 1 (page 4).

POINT 5: Why gender and race are not adjusted in the model?

This study is an ecological time series, so the main unit of analysis (represented by a row of data) is the number of visits per day and not the individual person. So we could assess modification by such factors but it is not necessary to adjust for them. We did not have information on race in our data set and, based on previous literature, we were more interested in the modifying effects of age but not sex for which there is very little expectation of a difference.

POINT 6: Figure 2. The labels are missing for y-axis.

Labels have now been added.

POINT 7: Results: The 0-15 age group shows a significant association between temperature and ED visits. However, it may be due to that this age group has increased activity and high risk of injury during summer. If that, the temperature should not be the cause of those ED visits. The author needs to make a discussion in the manuscript.

We think this is a good point and it is highly likely that children may be more active on hot days. The temperature effect could therefore be more indirect rather than a physiological response. We have added some brief speculations on this in the discussion in the revised text as suggested by the reviewer (line 206 page 9).

POINT 8: Line 235. I think this result is reasonable because the effect of cold temperature is more common in low attitude areas.

Thanks for your interesting point raised - No response required.

Reviewer 3 Report

The paper analyzes the Emergency Department (ED) visits in relation to outdoor temperature and shows moderate but nonetheless significant associations between the two. I think this topic is of importance for public health and it is a step forward in this direction. The paper seems well sourced and the discussion is interesting. However, I have reservations about the methodology considered and I am afraid that, with slightly different choices, results could be significantly different.

Broad comments

  • My main criticism concerns the treatment of lags in the linear model. Why consider only averages of lags with natural splines while standard methods are not available to estimate distributed lags. In its present form, the applied analysis does not account for decreasing importance of higher lags. In other words, why not use the DLNM that is successfully applied in mortality and morbidity studies? I suspect results could be quite different (and more realistic) with the DLNM.
  • I don’t really understand the difference between heat and cold models since the temperature range is identical in both models. From what I understood it seems that the only thing that differs is the maximum lag considered. I think that to properly estimate heat and cold effects, the authors should either be separated between summer and winter (to better account for seasonal adaptation) or fit a DLNM which naturally estimates different lag structures (I strongly advise it).
  • The presentation of statistical methods is not always clear. In particular, how did the authors adjust the model for autocorrelation (p. 2 l. 78)? Was an ARMA fitted on residuals?
  • What does adjusting the model with or without threshold means at p. 5, l.142 ? I suggest improving the overall presentation of methods.
  • Twice the authors refer to heat waves, which is not particularly relevant since no effect of heat wave (i.e. temperature extremes) is estimated here. This is especially problematic at p. 10 l.205 since nothing allows saying that there is and additional effect of heat waves here. Indeed, the curves in Figure 2 do not particularly “hinge” at extreme temperatures, with perhaps the exception of the Fracture one in subfigure a).

Minor comments

  • Is the international classification of diseases used to create the different diagnostic categories? If so, the associated codes should be provided.
  • I might be missing something but why are % change different between part 1) of Table 2 and Figure 3?
  • Figure 2b: Cardiac conditions: there is a typo in the axis label
  • Some acronyms are not properly introduced. I spotted “A & E” and “IRR”.
  • The terms should be harmonized between “social problems” and “social conditions”.

Author Response

Dear Reviewer of the International Journal of Environmental Research and Public Health,

Many thanks for your feedback and comments on the article "The effects of temperature on London Accident and Emergency Department attendances: a time-series regression analysis".

Please find below my point-by-point responses to your comments:

Broad comments

POINT 1: My main criticism concerns the treatment of lags in the linear model. Why consider only averages of lags with natural splines while standard methods are not available to estimate distributed lags. In its present form, the applied analysis does not account for decreasing importance of higher lags. In other words, why not use the DLNM that is successfully applied in mortality and morbidity studies? I suspect results could be quite different (and more realistic) with the DLNM.

We have used averaged measures only to assess the functional form of the relationship, but for quantification purposes we have used a distributed lag model, i.e. separate lags. We also assessed for non-linearity in relationships and modelled appropriately were there was evidence of such. DLNM models simply use cross-basis functions to model possible non-linear and delayed effects of exposure, as we have done here. Previous work has shown the consistency of results {Armstrong B, Models for the relationship between ambient temperature and daily mortality. Epidemiology 2006;17:624–31}. Rather than being more realistic, results from heavily parameterised models such as DLNMs have been criticised – see for example article below based on the following citation: {Longden T, The impact of temperature on mortality across different time zones. Climatic Change 2019;157:221-42}. 

https://theconversation.com/heat-kills-we-need-consistency-in-the-way-we-measure-these-deaths-120500

POINT 2: I don’t really understand the difference between heat and cold models since the temperature range is identical in both models. From what I understood it seems that the only thing that differs is the maximum lag considered. I think that to properly estimate heat and cold effects, the authors should either be separated between summer and winter (to better account for seasonal adaptation) or fit a DLNM which naturally estimates different lag structures (I strongly advise it).

Apologies, you are correct. The descriptions should be short lag and long lag, rather than heat and cold models. This has been changed (line 143, page 5).

POINT 3: The presentation of statistical methods is not always clear. In particular, how did the authors adjust the model for autocorrelation (p. 2 l. 78)? Was an ARMA fitted on residuals?

To detect and adjust for autocorrelation in this study model diagnostic tests were used such as the minimization of the partial autocorrelation function (PACF) by checking the correlogram visually. It estimates the degree of correlation between proximate days, having controlled for other proximate days. On checking the residuals of the full model, some autocorrelation remained at day one which was adjusted for by adding the 1-day lagged residual into the full model as an explanatory term.

We initially omitted this in the method section in the interest of keeping the article short but this has now been amended to provide a better understanding of the statistical methods (page 3, line 100 + Supplementary material Figure S3).

POINT 4: What does adjusting the model with or without threshold means at p. 5, l.142 ? I suggest improving the overall presentation of methods.

This refers to the steps taken prior to choosing the core model for the fracture outcome ie. before starting the quantification regression analysis. We formally tested the two options for evidence of non-linearity by comparing the Akaike Information Criterion (AIC - measure of model fit)

-model 1: modelled with the heat effect using a linear term

-model 2: heat effect modelled with a linear-threshold term (16 degrees Celsius) in it.

--> Model 2 had a lower AIC and was therefore chosen for the final regression analysis.

We agree this is not clearly explained in the text and has been amended in the method section (line 111-117, page 3).

Point 5:Twice the authors refer to heat waves, which is not particularly relevant since no effect of heat wave (i.e. temperature extremes) is estimated here. This is especially problematic at p. 10 l.205 since nothing allows saying that there is and additional effect of heat waves here. Indeed, the curves in Figure 2 do not particularly “hinge” at extreme temperatures, with perhaps the exception of the Fracture one in subfigure a).

Agreed, this was an oversight. We are not assessing heat-wave episodes.

Minor comments

  • Is the international classification of diseases used to create the different diagnostic categories? If so, the associated codes should be provided.

Unfortunately ICD codes are not used for categorizing A&E data and are therefore not provided in this study. The different diagnostic categories have been set by NHS digital which is a database containing details of all admissions, Accident and Emergency attendances at NHS hospitals in England. Initially this data is collected during a patient's time at hospital as part of the Commissioning Data Set. This is submitted to NHS Digital for processing and is returned to healthcare providers as the secondary uses services data set and includes information relating to payment for activity undertaken. It allows hospitals to be paid for the care they deliver. This same data can also be processed and used for non-clinical purposes, such as research and planning health services. Because these uses are not to do with direct patient care, they are called 'secondary uses'.

  • I might be missing something but why are % change different between part 1) of Table 2 and Figure 3?

Apologies as the initial graph was showing the results from the relative risk (RR) given by the regression model. For ease of interpretation, the effect estimates were converted and expressed as percentage change in daily patient volume (%). This measure of association is a conversion of the relative risk obtained from generilised linear model and represents the increase in the number of daily A&E visits per change in degree Celsius. This has now been amended to both show the same measure of effect.

  • Figure 2b: Cardiac conditions: there is a typo in the axis label

Labels have now been amended in the Figure.

  • Some acronyms are not properly introduced. I spotted “A & E” and “IRR

All ED acronyms have been changed to A&E (Accident and Emergency) as introduced in the beginning and IRR was also removed to avoid confusion.

  • The terms should be harmonized between “social problems” and “social conditions”.

This has now been harmonized to social conditions, many thanks.

Round 2

Reviewer 1 Report

Dear editor,

The authors of the article performed a reconfiguration of the manuscript, attending and justifying all the notes and suggestions made.
In view of this reorganization, the article is of good quality for approval by this reviewer.
Thanks.

Author Response

Dear Reviewer,

Many thanks for your time and comments.

Reviewer 3 Report

I am thankful to the authors for clarifying some points I had not understood well in the previous version of the manuscript. I think the new version is already much easier to read. There are still nonetheless a few cloudy points for me that I would like to discuss below.

Further discussion on previous points

  • Point 1. I am sorry to come back on the matter of lags but I am not sure I understand what is being done here. If I understand correctly, the form of the dose-response curve is being estimated using the average temperature of past 3 or 21 days. And after that, a (linear) DLM is applied.

    It doesn’t bother me that much for lag 2 actually, but I think it is important to do it right for lag 21. Attributing the same weight to all lags (as being done here) or attributing different weights (as in a DLM) when 21 days are considered, could significantly affect the curves observed in Figure 2b, and thus the following steps in the analysis. And it is hard to believe for me that the former is a better predictive model. The study of Armstrong mentioned in the response quite clearly suggests it.

    If overfitting worries the authors, some recent developments can be accounted for, see Gasparrini, A., Scheipl, F., Armstrong, B., Kenward, M.G., 2017. A penalized framework for distributed lag non-linear models. Biom 73, 938–948.
    Note that the use of very long lags for cold can also be seen as a form of overfitting (see e.g. Kinney, P.L., Schwartz, J., Pascal, M., Petkova, E., Le Tertre, A., Medina, S., Vautard, R., 2015. Winter season mortality: will climate warming bring benefits? Environmental Research Letters 10, 064016).

    I was also aware of the (very nice) study by Longden but, unless I am mistaken, its criticisms rather target the definition of the reference temperature, which has little to do with the number of parameters in the model. Thus, it has little relevance for the matter of lagging.

    I only want to be convinced that what the authors did is a relevant way to analyze their (important) data considering the tools available today. A simple AIC comparison could do the job for me. It is simply a matter of confidence in the results.

    Finally, I am not sure where the unconstrained DLM results are displayed. Is it in Table 2? I suspect that what is being displayed here corresponds to the cumulative association, and if so it should be stated more clearly.

  • Point 4. Thank you for clarifying this point, I better understand now and I think it is a good idea to introduce a threshold effect. I only think the threshold choice (16 °C)is a bit arbitrary, but I won’t necessarily request that it is amended in the manuscript. For the record, I would nonetheless point out that many clever threshold models exist in the statistics literature that could be of use to the authors for future studies. I would cite notably the following: Muggeo, V.M.R., 2003. Estimating regression models with unknown break-points. Statistics in Medicine 22, 3055–3071. https://doi.org/10.1002/sim.1545

New minor comments

  • Thank you for clarifying A&E in the manuscript. However, I realize now that it is still not completely clear to me what it represents. I would suggest defining it a bit in the data section. For instance, do people dying after being admitted count in A&E?
  • Please arrange the order of supplementary Figures (S3 appears before S1 in the text).
  • At line 128 (p. 3), PM10 is not written as usual.

Author Response

Dear Reviewer,

Please find my point-by-point response uploaded as an attachment.

Many thanks for your time and comments.
